

# Nonlocal order parameter of pair superfluids

**Nitya Cuzzuol★, Luca Barbiero and Arianna Montorsi†**

Institute for Condensed Matter Physics and Complex Systems,
DISAT, Politecnico di Torino, I-10129 Torino, Italy

★ nitya.cuzzuol@polito.it , † arianna.montorsi@polito.it

## Abstract

Order parameters represent a fundamental resource to characterize quantum matter. We show that pair superfluids can be rigorously defined in terms of a nonlocal order parameter, named odd parity, which derivation is experimentally accessible by local density measurements. As a case of study, we first investigate a constrained Bose-Hubbard model at different densities, both in one and two spatial dimensions. Here, our analysis finds pair superfluidity for relatively strong attractive interactions. The odd parity operator acts as the unique order parameter for such phase irrespectively to the density of the system and its dimensionality in regimes of total particle number conservation. In order to enforce our finding, we confirm the generality of our approach also on a two-component Bose-Hubbard Hamiltonian, which experimental realization represents a timely topic in ultracold atomic systems. Our results shed new light on the role of correlated density fluctuations in pair superfluids. In addition, they provide a powerful tool for the experimental detection of such exotic phases and the characterization of their transition to the atomic superfluid phase.

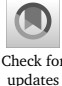
# 1 Introduction

Quantumness is fundamentally rooted in delocalization and entanglement, the latter allowing quantum matter to organize in low temperature phases which escape Landau classification of spontaneously symmetry breaking (SSB) phases [1, 2]. Notable examples are topological phases [3–7], as well as insulators with continuum symmetries, i.e. low dimensional Mott insulators (MI) [8]. Crucially, while SSB phases are endowed with a local order, the aforementioned examples are characterized by nonlocal orderings which do not violate the Mermin-Wagner theorem [9]. Such intrinsic nonlocality allows for insulators with continuous symmetries both in 1D and 2D to be accurately captured by nonlocal parity operators [10–14], while nonlocal string order parameters [15–17] result the fundamental quantity to capture one dimensional topological insulators. Relevantly, from SSB [18–24] to low dimensional Mott [25–28] and topological [29, 30] insulators, experimental setups working with ultracold bosonic atoms in optical lattices [31–34] have allowed for an in depth description of such states of quantum matter. Besides the impressive versatility encoded in such quantum simulators, a game changing aspect is represented by highly accurate probing schemes. By indeed making use of either noise correlators [35, 36] or quantum gas microscopy [37] both local and nonlocal orderings have been efficiently probed.

Although the above discussion focused on insulating states of matter, the notion of nonlocal orders can be applied also to characterize conducting phases of low-dimensional interacting fermions [12, 38–40]. In this regard, nonlocal parity [11, 12] and string [17] operators relative to the spin degrees of freedom have been efficiently employed to characterize Luther Emery liquid [11] and superconducting phases [38–40] as well as topological conducting regimes [38, 40] respectively. At the same time, the order parameters of gapped conducting phases in low dimensional bosonic matter are still unexplored. In this paper we tackle this important topic.

In particular, we explore conducting phases characterized by bosonic pairing, i. e. pair superfluids (PSFs) whose existence has been predicted in large variety of physical systems [41–54] and corresponds to the XY2 phase in spin models [44, 55, 56]. Importantly, PSFs are neither associated to a symmetry breaking nor to the appearance of topological order, thus representing an example of a gapped superfluid with continuous symmetry. To perform our analysis, we consider a paradigmatic model of strongly interacting bosons, namely the Bose-Hubbard model with a constraint of a maximum of two bosons per site. It was initially introduced to describe regimes characterized by a large rate of three-body losses in ultracold atomic gases [41, 42]. Previous theoretical works [41, 43] have established that for weakly attractive interaction an atomic superfluid (SF) phase occurs while, by increasing the attraction between bosons, the system undergoes a phase transition. Here, a different SF phase, i.e. the PSF where pairs of single bosons become the effective elementary constituents, appears. As we will discuss in details, these different superfluids have been characterized by analyzing the different long distance decay of the single particle and pair-pair correlation function [41]. Although certainly useful, this approach poses significant challenges. On one hand it makes difficult to properly locate the transition point. On the other hand, experimental schemes to probe the long distance behavior of correlators are still characterized by a limited efficiency. Here, we propose an alternative approach based on local density measurements, which can be performed with high accuracy through quantum gas microscopy.

Specifically, we exploit the bosonization approach [8, 10, 57] to these system to look for the nonlocal order parameter which corresponds to the PSF phase in one dimension. In so doing, we introduce a new nonlocal parameter, named **odd parity**. The prediction is then verified also numerically, by extensive DMRG numerical simulation, confirming that the odd parity is the unique order parameter of the PSF phase, in regimes where the total particle

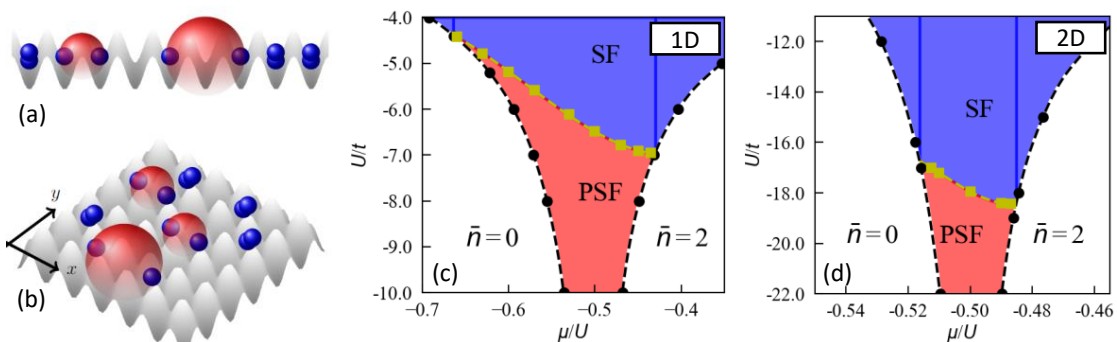

Figure 1: Cartoon of a phase with finite odd parity order (see (8) and (10)) for a 1D (a) and 2D (b) lattice respectively. It amounts to a background of holons (empty sites) and doublons (doubly occupied sites) in which density fluctuations (single bosons) occur only in pairs with finite correlation length, highlighted by red shaded regions. Panels (c) and (d) present the resulting phase diagrams obtained through iDMRG simulations for the 1D system (with $\chi_{\max} = 200$) and a lattice system with $M = 6$ (with $\chi_{\max} = 1500$), respectively. The SF region (depicted in blue) corresponds to zero expectation value of odd parity, while the PSF region (depicted in red) is characterized by a non-zero value.

number is conserved. As PSF can occur also in 2D [43, 45, 46, 48], we extend the definition of odd parity by means of the brane generalization [39, 58] and we verify numerically that our approach holds also beyond the one dimensional limit. In addition, since sample imperfections might make it challenging to accurately control the particle density, we show that our method remains highly robust in a large range of bosonic fillings. Finally, we test the validity of our predictions on an experimentally feasible two component Bose-Hubbard Hamiltonian where the SF-PSF phase transition was recently predicted [59–64].

## 2 The model

We analyse the Bose-Hubbard model corresponding to the following Hamiltonian

$$H = -t \sum_{\langle R,R' \rangle} (b_R^\dagger b_{R'} + \text{h.c.}) + \frac{U}{2} \sum_R n_R(n_R - 1) + \mu \sum_R n_R, \tag{1}$$

where $R = (x, y)$ is a generic site of a lattice composed by $L$ rungs and $M$ legs ($1 \le x \le L$; $1 \le y \le M$, $M = 1$ in 1D) schematized in figure (1)(a,b). Here, $\langle R, R' \rangle$ indicates nearest-neighbor sites and $b_R$, $b_R^\dagger$ are the bosonic creation and annihilation operators on the site $R$. In addition, $\mu$ is the system chemical potential and the hopping matrix element $t = 1$ fixes the energy scale. In order to favor the bosonic pairing we focus on regimes with on-site attractive interaction $U < 0$ and we impose the three body constraint $(b_R^\dagger)^3 = 0$ which allows to prevent the system collapse. As shown both in the 1D and 2D case in figure (1)(c,d), bosons remain in the SF phase up to a critical value of the attractive interaction, where PSF emerges. To locate the SF-PSF transition one can investigate the long distance behavior of the single particle and pair-pair correlation functions defined as

$$S(r) = \langle b_{x,y}^\dagger b_{x+r,y} \rangle, \tag{2}$$

$$D(r) = \langle (b^\dagger)_{x,y}^2 (b)_{x+r,y}^2 \rangle, \tag{3}$$

respectively.[1] In fact, one expects that in the SF phase both correlators exhibit algebraic decay, signaling the presence of quasi-long-range order. In contrast, in the PSF phase, one can argue that while $S(r)$ exhibits exponential decay, indicating the suppression of single bosons motion, $D(r)$ should maintain its polynomial decay [41].

These intuitive aspects reflect the specific behavior of two different energy gaps defined as:

$$\Delta_{\alpha=1,2}(L \times M) = E(N+\alpha; L \times M) + E(N-\alpha; L \times M) - 2E(N; L \times M), \tag{4}$$

where $E(N; L \times M)$ is the ground state energy of a system of $N$ particles on $L \times M$ sites. Specifically, one can find that, for a finite size system, $\Delta_2 > \Delta_1$ in SF while $\Delta_2 < \Delta_1$ in PSF [44]. On the other hand, in the thermodynamic limit one finds that $\Delta_2 = 0$ for any value of $U$, while $\Delta_1 > 0$ when entering the PSF regime and thus explaining the exponential decay of $S(r)$.

## 2.1 Bosonization analysis

The 3-body constraint Bose-Hubbard model at filling one is directly connected to spin-1 XXZ chain with single ion anisotropy [10, 44, 56, 65, 66]. This aspect is at the core of the analytical treatment of (1) in 1D by means of a bosonization approach [8, 57]. Specifically, by following [10, 55], the Hamiltonian (1) is first mapped into a spin-1 model, then each spin-1 variable is written as a sum of two spin-1/2, which in turn are mapped onto two spinless fermions via a Jordan Wigner transformation. The two corresponding fermionic ladder operators are bosonized in the standard way, introducing two bosonic fields $\phi_{1,2}(x)$, and their dual fields $\theta_{1,2}(x)$ such that $[\phi_\alpha(x), \theta_\beta(y)] = i\pi\delta_{\alpha\beta}\Theta(x-y)$, with $\alpha, \beta = 1, 2$ and $\Theta(x-y)$ being the Heaviside step function. Apart from a coupling term negligible in the low energy limit, the resulting Hamiltonian decouples when re-written in terms of the symmetric an anti-symmetric combinations of $\phi_{1,2}(x)$, namely $\phi_\pm(x) = \phi_1(x) \pm \phi_2(x)$ and $\theta_\pm(x) = (\theta_1(x) \pm \theta_2(x))/2$. Explicitly

$$H \rightarrow H_+ + H_-, \tag{5}$$

with

$$H_+ \equiv \frac{u_+}{2\pi} \int dx \left[ K_+(\partial_x \theta_+)^2 + \frac{1}{K_+}(\partial_x \phi_+)^2 \right] + \frac{g}{(\pi a)^2} \int dx \cos(2\phi_+), \tag{6}$$

$$H_- \equiv \frac{u_-}{2\pi} \int dx \left[ K_-(\partial_x \theta_-)^2 + \frac{1}{K_-}(\partial_x \phi_-)^2 \right] - \frac{g}{(\pi a)^2} \int dx \cos(2\phi_-) - \frac{t}{\pi a} \int dx \cos(2\theta_-). \tag{7}$$

Here $a$ is the lattice spacing, $g = -aU/2$ is the coupling constant, and $K_\pm = 2(\sqrt{1 \pm U/t\pi})^{-1}$ are the Luttinger parameters, while $u_\pm = 2ta/K_\pm$ are the velocities in the $\pm$-channels. We recognize in $H_\pm$ two sine-Gordon models where the condition $U < 0$ implies $K_+ > 2$ and $K_- < 2$. A renormalization group analysis [10] suggests that in this regime the field $\phi_+$ never pins, whereas $\phi_-$ (for $U < -2t\pi$) can pin to the value 0. The corresponding phase can be identified with the PSF phase.

On general grounds, the bosonization approach can also be exploited to associate to each pinning value of the bosonic fields an appropriate lattice nonlocal operator which expectation value becomes different from zero in the corresponding phase. This was first proposed in the framework of spin-1 chains in [67]. For fermionic models, this was done in [11, 12, 17]. More specifically, in the bosonic case, it was shown that the MI phase of the 1D Hamiltonian (1) can be characterized by the non vanishing of the expectation value of the parity

---

[1]In (2) and (3), the $y$-th leg in the lattice is constant but can assume any value. However, since the model is isotropic in both hopping and interaction, any $y$-th leg leads to the same effects. In one dimension, obviously $y = 1$.

operator [10], namely $O_{\mathrm{P}}(j) \equiv \prod_{x=1}^{j-1} \exp[i\pi(n_x - 1)]$, which in the continuous limit becomes $O_{\mathrm{P}}(x) \sim \cos(\phi_+(x))$. In the thermodynamic limit, its expectation value will remain finite when most sites are singly occupied, and holon (empty sites) doublon (doubly occupies sites) fluctuations occur in correlated pairs.

Here we propose that when instead the field $\phi_-$ pins to the value 0 a different parity operator $O_{\mathrm{P}}^{(\mathrm{o})}$ should become finite, defined as

$$O_{\mathrm{P}}^{(\mathrm{o})}(j) = \prod_{x=1}^{j-1} \exp[i\pi n_x]. \tag{8}$$

We name it *odd parity*,[2] since its expectation value is finite when fluctuations occur in the form of correlated pairs of single bosons. Figure (1)(a) shows a simple cartoon of the case when this quantity is nonvanishing in 1D.

In fact, one can closely follow the derivation exploited in [11, 17] for the spin parity in the fermionic case to show how in the continuum limit the proposed non local operator (8) is connected to the bosonic field $\phi_-$. Explicitly, upon rewriting the product of exponentials as the exponential of a sum, the latter in the continuum (where $j$ is replaced by $r$) becomes the integral over $x$ of the density of the bosonic field $\phi_-$, which can be written as $\partial_x \phi_-$. The integral can then be straightforwardly performed, and the parity operator is identified with the symmetric form of $\exp[i\pi\phi_-(r)]$, namely

$$O_{\mathrm{P}}^{(\mathrm{o})}(r) \sim \cos\phi_-(r), \tag{9}$$

whose expectation value is finite when $\phi_-(r) = 0$. Thus the latter should be finite in the whole PSF phase, and vanishing as soon as $\phi_-$ unpins.

This corresponds to the presence of a uniform density distribution over the chain, in which holon-doublon density fluctuations may occur only in correlated pairs. Whereas in the PSF phase $\langle (O_{\mathrm{P}}^{(\mathrm{o})}(j) + O_{\mathrm{P}}^{(\mathrm{o})}(j+1))/2 \rangle$ will converge to a finite $O_{\mathrm{P}}^{(\mathrm{o})}$ while $O_{\mathrm{P}}$ is vanishing. This capture the prevalence of an holon-doublon distribution over the chain, in which single boson density fluctuations may occur only in correlated pairs, as shown in the cartoon of figure 1.

When extending this analysis towards the 2D limit, upon exploiting [13, 14, 58] one may introduce a brane odd parity to capture the PSF phase for an $M$ leg ladder. This is the product of $M$ odd parities for each $x$-value in the lattice. Explicitly

$$O_{\mathrm{P}}^{(\mathrm{o})}(j) = \prod_{x=1}^{j-1} \exp\left[i\pi \sum_{y=1}^{M} n_{x,y}\right], \tag{10}$$

whose expectation value is nonvanishing at finite $M$ in the thermodynamic limit when in the disordered background of holons and doublons density fluctuations occur only in correlated pairs, as shown schematically in figure (1)(b). In fact, based on previous results in similar cases [14, 26, 39, 68], we argue that its finiteness in the 2D limit $M \to \infty$ would be achieved upon proper rescaling of the argument of the exponential with $M$. This is consistent with extrapolating the 2D phase transition by direct inspection of its occurrence at finite $M$ through the unrescaled operator (10).

## 3 Numerical results

We present in this section the results we obtained by performing iDMRG [69–71] simulations, which allows to accurately take into account the quantumness of a given system. We analyse

---

[2]Since $O_{\mathrm{P}}^{\mathrm{o}}(j) = (-)^{j-1} O_{\mathrm{P}}(j)$, in the subsequent discussion it will be relevant.

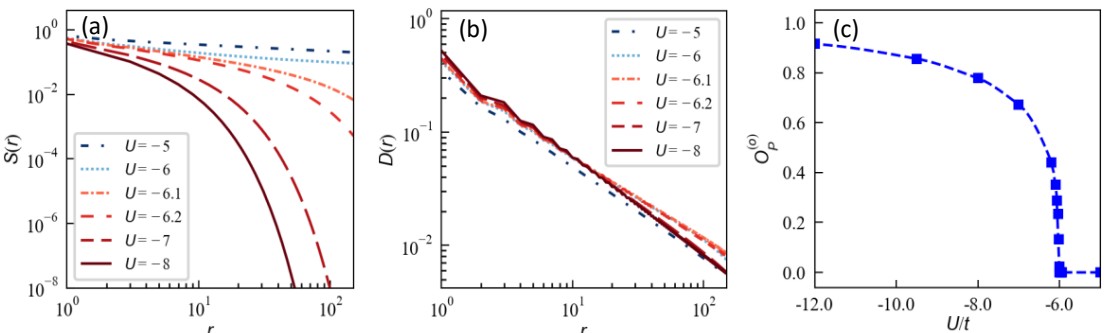

Figure 2: Analysis of a 1D system ($M = 1$) at $\bar{n} = 1$ by performing iDMRG simulations with a maximum bond dimension of $\chi_{\max} = 200$. The panels (a) and (b) present the analysis of the correlators in (2),(3) respectively. In the panel (c), the odd parity $O_{\mathrm{P}}^{(\mathrm{o})}$ is evaluated.

first the system of a single chain ($M = 1$) and then we extend our investigation to larger systems aiming to reach the 2D limit. Within this algorithm, we can either fix a priori the particle density, working in the canonical ensemble, or tune the chemical potential, in a grand canonical framework. In this regime, where the particle number conservation is released, $\langle b_i^2 \rangle$ is an additional order parameter of PSF [41].

Firstly, for both the 1D and 2D cases, we present a validation at fixed filling $\bar{n} = 1$ of the effectiveness of exploiting the odd parity in estimating the SF-PSF transition, in comparison to the use of correlators $S(r)$, $D(r)$ as defined in (2, 3). Then we present our resulting phase diagrams in $(\mu, U)$ plane, shown in figure (1)(c,d). Next, for both the 1D and 2D cases, we further investigate the filling dependence of the SF-PSF phase transition by imposing $\bar{n} \neq 1$ throughout the transition.

Referring to previous discussion about parity operators in section 2.1, the nonlocal order parameter $O_P^{(o)}$ has been extracted from $O_P^{(o)}(j)$ by fixing $j$ to a sufficiently large value to ensure that, within the region of parameter space where the model exhibits the phase defined by this parameter (i.e., not close to the transition), the nonlocal order parameter has already converged to a finite constant. We verified that in the same region the parameter $O_P(j)$ has an oscillating behaviour with vanishing average.

## 3.1  Results in 1D

Our investigation starts with the study of local correlators in (2) and (3) by fixing $\bar{n} = 1$ and enforcing particle number conservation. As illustrated in figure (2)(a) and (b) and mentioned before, we find that strong attractions generate an exponential decay of $S(r)$ while $D(r)$ maintains its quasi-long range order for any $U$. This behavior can be compared with that of the odd parity, as shown in figure (2)(c). Here, $O_{\mathrm{P}}^{(\mathrm{o})}$ shows a sharp and well defined transition point. Specifically, the odd parity becomes finite at a specific $U_{\mathrm{c}}$ which clearly does correspond to the one where the long-range behavior of $S(r)$ starts to deviate from an algebraic decay.

Our numerical approach makes it also possible to accurately complement this analysis by incorporating a study of the gaps $\Delta_{1,2}$. Since in this case we are interested in single excitations, we are forced to the finite DMRG and a thermodynamic extrapolation (for comparison, the calculation of the odd parity using both methods is detailed in Appendix A). In this limit, figure (3) shows indeed that in the SF $\Delta_1 = \Delta_2 = 0$ while, as long as PSF takes place, $\Delta_2 = 0$ but $\Delta_1$ becomes finite. Notably, the condition $\Delta_1 > 0$ turns out to be fulfilled exactly when $O_{\mathrm{P}}^{(\mathrm{o})} > 0$ therefore unambiguously proving the validity and accurateness of our approach.

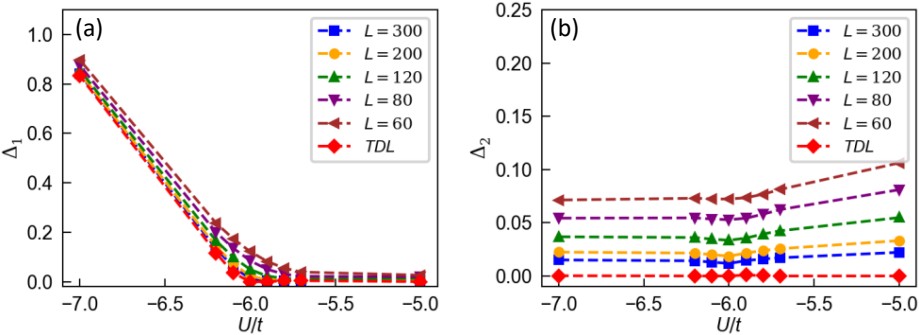

Figure 3: Analysis of the gaps defined in (4) by increasing the size of the chain $L = 60, 80, 120, 200, 300$ for finite DMRG. One can see in (a) the single particle gap while in (b) the double particle gap. Only the first one remains finite by extrapolating the thermodynamic limit (TDL) by considering a second order polynomial in $1/L$. We keep as maximum bond dimension $\chi_{\max} = 200$.

In order to further enforce our results, restricted so far to the canonical ensemble, we also inspect the odd parity in the grand canonical ensemble by tuning both $U$ and $\mu$. The resulting phase diagram is reported in figure (1)(c), based on the analysis of $O_{\mathrm{p}}^{(\mathrm{o})}$. It confirms the presence of the PSF phase (in red), in which the odd parity order remains robust irrespectively to the particle density.

## 3.2 Results in 2D

The simulation of 2D systems using MPS algorithms can be challenging due to the efficiency of this method relying on covering a finite ratio of the system's entanglement entropy. However, the entanglement entropy grows extensively according to the area law, diminishing the algorithm's efficiency [71, 72]. In our iDMRG simulations, we address this challenge by considering an infinite size in the $x$-direction, achieved through the repetition of a finite unit cell in the $y$-direction. The use of periodic boundary conditions accelerates convergence to a 2D configuration. The resulting infinite cylinder system can be effectively mapped to a 1D system by following a 'snake-like' path [70], initially traversing in the $y$-direction.

We start again from the canonical ensemble by considering the different behaviours of the correlators (2) and (3) at $\bar{n} = 1$. As in the previous case, figure (4)(a) and (b) shows that for a sufficiently negative $U$ there is a change in the decay law for $S(r)$ while $D(r)$ conserves his long distance behavior. As before, this point signals the emergence of the PSF phase. Again, we show in figure (4)(c) this transition to coincide with the critical $U_{\mathrm{c}}$ where the odd parity becomes finite.

Then, by increasing $M$ from 3 to 8, we evaluated the odd parity to extrapolate the critical interaction for SF-PSF transition in the 2D limit,[3] as illustrated in figure (4)(c).

For $M = 6$ we further release the particle number conservation and evaluate again the odd parity by varying the onsite attractive interaction $U$ in the $(\mu, U)$ plane. The resulting phase diagram, which should be representative of the 2D limit, is shown in figure (1)(d): also in this case the presence of the PSF phase (in red) is in one-to-one correspondence with the region of finite odd parity. This is confined between the SF phase (in blue), and the two trivial phases (in white) corresponding to the $\bar{n} = 0$ and $\bar{n} = 2$ cases.

---

[3]We consider the case $M = 6$ as a reliable approximation of a 2D system, given that the difference between this case and our resulting limit is less than 3%.

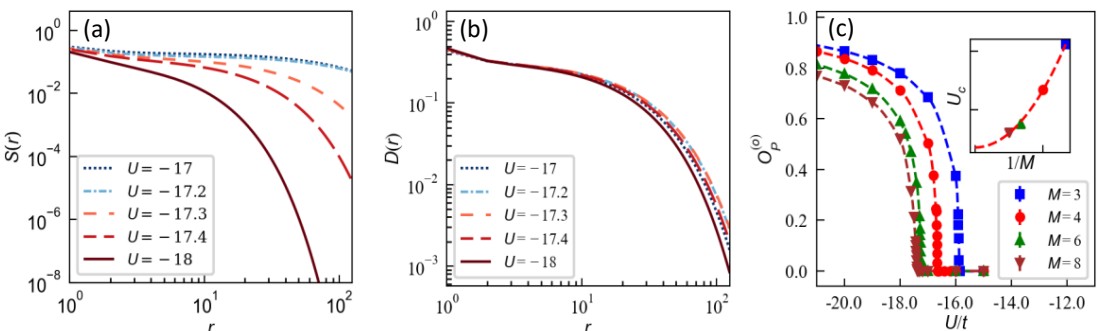

Figure 4: Analysis for a 2D system by iDMRG simulations. The first two panels (a) and (b) present the correlators in (2),(3) respectively, by increasing the onsite attractive interaction $U < 0$ for $M = 6$. Panel (c) presents the plot of the SF-PSF transition for 2D systems as a function of the number of legs $M$. The inset shows the extrapolation for a square 2D lattice of the critical interaction $U_c(2D) = -17.65 \pm 0.05$ imposing $\chi_{max} = 1500$. Similar to the 1D case, a comparison of the three plots reveals that, coinciding with the change in decay observed in $S(r)$ the odd parity becomes finite.

## 3.3 Arbitrary filling

The previously discussed results for 1D and 2D systems focused first on the canonical ensemble, where we enforced a filling of $\bar{n} = 1$. Subsequently, such condition was released upon exploring the behavior of the system in the grand canonical ensemble, where the number of bosons is not necessarily constant throughout the SF-PSF phase transitions, and we obtained the $(\mu, U)$ 1D and 2D phase diagrams. To confirm the picture, we now inspect the SF-PSF transition again in the canonical ensemble, for fixed fillings $\bar{n} \neq 1$.

The results are shown in figure (5) for a 1D (a) and a 2D (b) system. There the odd parity (8) was evaluated for fillings greater and smaller than one. The SF and PSF phases are again distinguished through the behavior of odd parity: in the first, the odd parity is zero, whereas in the other the emergence of a nonlocal order is confirmed by its non zero value. A careful analysis reveals how at all fillings the transition is second order, in both 1D and 2D cases.[4] Moreover, in this way the actual dependence of the critical value of interaction $U_c$ on filling is obtained.

## 3.4 Two-species system

The presence of the SF-PSF phase transition we discussed in previous sections is not limited to the 3-body constrained BHM. For instance, it is observed also in case of two components hardcore bosons in optical lattice [59–64]. Although, the regime of PSF has not yet been observed, Bose mixtures in optical lattice have been the subject of intense experimental activity [73–80]. The Hamiltonian we are interested in reads

$$H = -t \sum_{i,\sigma=A,B} (b^\dagger_{i,\sigma} b_{i+1,\sigma} + \text{h.c.}) + U \sum_i n_{i,A} n_{i,B}, \tag{11}$$

where $i$ denotes the generic site of each chain for boson species $\sigma = A, B$. For attractive interchain on-site interaction, the system should show a tendency to create pairs of A-B bosons.

---

[4]This is at variance with the case of fixed chemical potential, in which traversing a transition in the $(\mu, U)$ plane may result in the total number of bosons not being conserved, leading to abrupt filling changes. This discontinuity can also manifest in odd parity, transitioning from zero to a finite value.

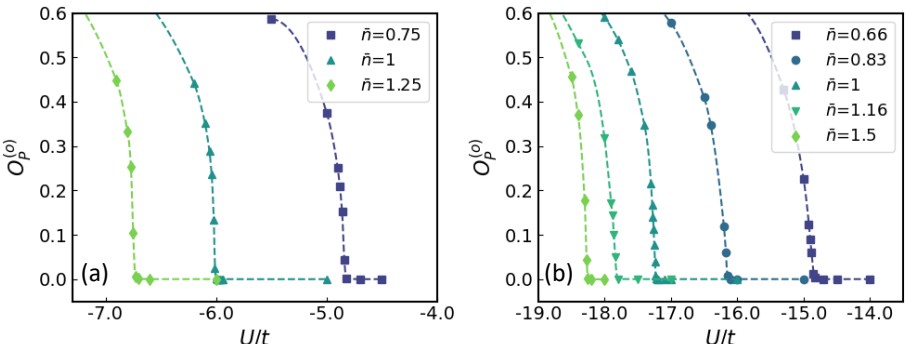

Figure 5:   Evaluation of the odd parity $O_\mathrm{P}^{(\mathrm{o})}$ for a fixed $\bar{n} \neq 1$ for the 1D system in (a) and 2D in (b), both by performing iDMRG simulations for $\chi_\mathrm{max} = 200, 1500$ respectively.  The $\bar{n} = 1$ case is included for comparison with previous results.  As evident from figures, for the considered densities there is no change in the nature of the transition when moving away from filling one.

As before we can measure the emergence of the odd parity order. The previous definition (8) in this case becomes

$$O_\mathrm{P}^{(\mathrm{o})}(j) = \prod_{i=1}^{j-1} \exp\left[ \sum_{\sigma=\mathrm{A,B}} i\pi n_{i,\sigma} \right]. \tag{12}$$

Now the same site on the two chains is either empty or occupied in both chains, whereas the fluctuations occur in the form of correlated pairs of bosons of the different species as depicted in figure (6)(a).

In agreement with results in [64, 81], and with bosonization analysis in the fermionic case [11], $O_\mathrm{P}^{(\mathrm{o})}$ becomes finite as soon as $U < 0$, as shown in figure (6)(b).

## 4   Conclusions

We unveiled the order parameter of the exotic pair superfluid phase. The latter takes the form of a nonlocal parity operator named odd parity which can be accurately detected through local density measurements. Such quantity is finite whenever in a disordered background of holons and doublons single bosons occur in correlated pairs. The prediction is confirmed by a bosonization analysis and extensive numerical simulations.

Relevantly, we have shown our results to hold both in one and two spatial dimensions, at any density supporting a pair superfluid phase, and in two different models where such state of matter takes place. This aspect confirms the generality and the large range of applicability of our findings.  Moreover, we proved the aforementioned points through extensive matrix-product-state simulations where quantum fluctuation are accurately taken into account. This makes our results highly robust and reliable in deep quantum regimes where mean-field approaches are not reliable.

We further stress that a major advantage of nonlocal operators is that they do not break continuous Hamiltonian symmetries, thus describing orders that can persist in low dimensional systems even at non vanishing temperatures. Hence the measurement of odd parity order is even more at hand in present experimental setups, especially because it can be obtained by means of just local density probes.  In conclusion, we believe that our results represent an important step towards a more complete and accurate characterization of strongly correlated many-body quantum systems.

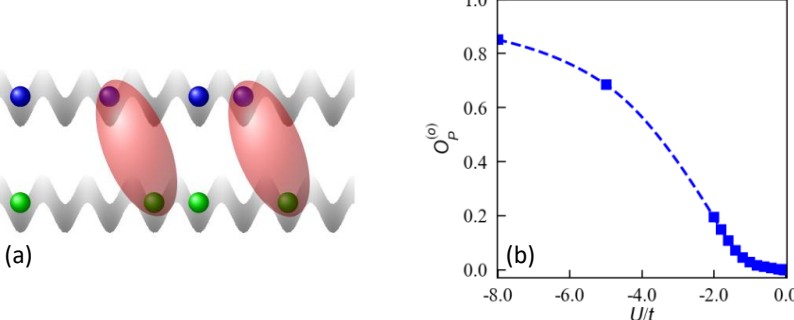

Figure 6: (a) Schematic representation of the two hard-core bosons chains model. One can see the direct connection to the previous representation in figure (1)(a). Now the doublons are composed by A (in blue) and B (in green) bosons with the same site index. For finite odd parity, such doublons can be decoupled and differ by one (or more) lattice spacing, while maintaining a finite correlation length highlighted by the red ellipses. (b) Evaluation of the odd parity in (12) for SF-PSF phase transition performing iDMRG simulations with a maximum bond dimension of $\chi_{\max} = 400$. Differently from previous case, the transition occurs as soon as the interaction $U$ becomes negative.

## Acknowledgments

Calculations were performed using the TeNPy Library (version 0.10.0) [70].

**Funding information** N.C. and A.M. thank for the financial support from the ICSC – Centro Nazionale di Ricerca in High Performance Computing, Big Data and Quantum Computing, funded by European Union – NextGenerationEU (Grant number CN00000013). L. B. acknowledges financial support within the DiQut Grant No.2022523NA7 funded by European Union – Next Generation EU, PRIN 2022 program (D.D. 104 - 02/02/2022 Ministero dell'Università e della Ricerca). Computational resources were provided by HPC@POLITO (http://www.hpc.polito.it).

## A   Comparison of Odd Parity calculations: finite vs infinite DMRG

As introduced in the text, we exploited two different variants of the DMRG algorithm for our numerical simulations [71, 82–85] in matrix product states formalism [71, 86–88]. One is the DMRG for a finite system. We used it for computing the energy gaps and requires an infinite size extrapolation, as we did in figure (3). The other one is the infinite-size variant of the DMRG (iDMRG) algorithm [69, 71, 89, 90], and this is the one we exploited to compute the odd parity (as specified in section 3) in figures (1,2,4-6). It leads to obtain a fixed-point translationally invariant matrix product wavefunction. When the algorithm is converged, the resulting quantum state approximates the 1D thermodynamic limit. In this way we are able to perform our simulations directly for an approximated infinite chain, avoiding the necessity of an infinite size extrapolation typical of the finite size DMRG results. However, in practice, the simulated chain is not infinite and we are forced to select a finite maximum $j$ in accordance with the algorithm, when computing the expectation of the odd parity operator (as in eq.(8)). Not too close to the transition, $O_P^{(o)}$ quickly converges to zero or a constant, and thus a small value of $j$ is sufficient to have a perfectly converged $O_P^{(o)}$.

Contrary to the calculation of $O_P^{(o)}$, extracting the gaps requires a finite chain and so finite size simulations. Indeed the iDMRG method doesn't support the calculation of excitations and one has to extrapolate the thermodynamic limit of the systems from a set of finite sizes.

To assess the consistency between the two procedures, we can compare the results from both algorithm for $O_P^{(o)}$ of the 1D Bose-Hubbard model in eq.(1). This is shown in figure (7). The light blue crosses are the iDMRG results (the same as in figure (2).(c)), while the red diamonds are the extrapolated results from finite size DMRG. Both methods distinguish between a region of zero $O_P^{(o)}$ (the atomic SF) and one where the same quantity is finite (PSF): the estimated transition point by iDMRG is in accordance with the extrapolated TDL data. This comparison shows that, by exploiting the iDMRG method, we are obtaining an accurate approximation of the TDL, avoiding the extrapolation procedure. Moreover the match between iDMRG and finite size results is supported by the agreement with the opening of the single particle gap $\Delta_1$ in figure (3).

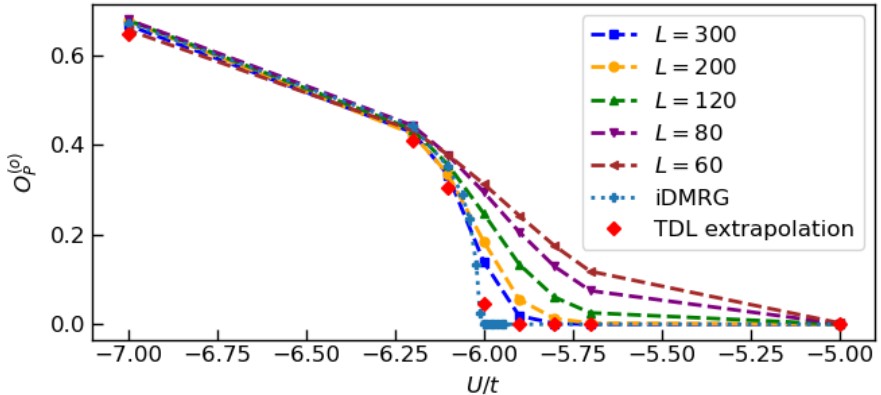

Figure 7: The figure shows the $O_P^{(o)}$ results for: different sizes of a finite chain by DMRG, the thermodynamic limit extrapolation obtained by these sizes, and the same quantity obtained by the iDMRG method. The dashed and dotted lines are guide for the eyes.

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
