# Peer review of "Nonlocal order parameter of pair superfluids"

_SciPost Physics, doi:SciPost Phys. Core 8, 020 (2025)_

## Round 2 · Referee Report · Anonymous (Referee 1) · 2024-6-2

Strengths

The paper by Cuzzuol et al. is a vry interesting paper a kind if exotic superfluids, the so calls air sulerfluids. The authors introduce a novel nonlocal order parameter, named odd parity. They show that pair superfluids can be rigorously defined in terms of it.Moreover , this order parameter is experimentally accessible. As a case of study, they investigate a constrained Bose-Hubbard model at different densities, both in one and two spatial dimensions. Here, pair superfluidity occurs for relatively strong attractive interactions. The odd parity operator acts as the unique order parameter for such a phase. They apply also their approach to a two-component Bose-Hubbard Hamiltonian. Such a system is at the centre of interestinf investigations in ultracold atoms.

The paper is very clear and very well written. The studies are very profound and cover all relevant aspects of the problem. I recommend publication in SciPOst.

I have only minor suggestions:
a)it would be useful to core a review on non-standard Hubbardmodels where pair supefluidity is discussed thoroughly in various context:

1. Omjyoti Dutta, Mariusz Gajda, Philipp Hauke, Maciej Lewenstein, Dirk-Sören Lühmann, Boris A. Malomed, Tomasz Sowiński, Jakub Zakrzewski, Non-standard Hubbard models in optical lattices, Rep. Prog. Phys. 78, 066001 (2015), arXiv:1406.0181.
2. Titas Chanda, Luca Barbiero, Maciej Lewenstein, Manfred J. Mark, and Jakub Zakrzewski, Recent progress on quantum simulations of non-standard Bose-Hubbard modes, arXiv:2405.07775

Weaknesses

no weaknesses

Report

The paper by Cuzzuol et al. is a vry interesting paper a kind if exotic superfluids, the so calls air sulerfluids. The authors introduce a novel nonlocal order parameter, named odd parity. They show that pair superfluids can be rigorously defined in terms of it.Moreover , this order parameter is experimentally accessible. As a case of study, they investigate a constrained Bose-Hubbard model at different densities, both in one and two spatial dimensions. Here, pair superfluidity occurs for relatively strong attractive interactions. The odd parity operator acts as the unique order parameter for such a phase. They apply also their approach to a two-component Bose-Hubbard Hamiltonian. Such a system is at the centre of interestinf investigations in ultracold atoms.

The paper is very clear and very well written. The studies are very profound and cover all relevant aspects of the problem. I recommend publication in SciPOst.

I have only minor suggestions:

Requested changes

Minor suggestions

a)it would be useful to core a review on non-standard Hubbardmodels where pair supefluidity is discussed thoroughly in various context:

1. Omjyoti Dutta, Mariusz Gajda, Philipp Hauke, Maciej Lewenstein, Dirk-Sören Lühmann, Boris A. Malomed, Tomasz Sowiński, Jakub Zakrzewski, Non-standard Hubbard models in optical lattices, Rep. Prog. Phys. 78, 066001 (2015), arXiv:1406.0181.
2. Titas Chanda, Luca Barbiero, Maciej Lewenstein, Manfred J. Mark, and Jakub Zakrzewski, Recent progress on quantum simulations of non-standard Bose-Hubbard modes, arXiv:2405.07775

Recommendation

Publish (surpasses expectations and criteria for this Journal; among top 10%)

  • validity: top
  • significance: top
  • originality: top
  • clarity: top
  • formatting: perfect
  • grammar: excellent

Author:  Nitya Cuzzuol  on 2024-06-20  [id 4577]

(in reply to Report 1 on 2024-06-02)

Dear Referee,

Thank you very much for your valuable suggestions and comments. We appreciate the time and effort you have put into reviewing our paper. We will carefully consider your feedback and incorporate the suggested changes into our manuscript.

Best regards,
Nitya Cuzzuol

---

## Round 2 · Referee Report · Anonymous (Referee 2) · 2024-6-29

Report

Warnings issued while processing user-supplied markup:

  • Inconsistency: plain/Markdown and reStructuredText syntaxes are mixed. Markdown will be used.
    Add "#coerce:reST" or "#coerce:plain" as the first line of your text to force reStructuredText or no markup.
    You may also contact the helpdesk if the formatting is incorrect and you are unable to edit your text.

Dear Editors,

This is my reviewer's report on the manuscript https://scipost.org/submissions/2404.15972v2/, by N. Cuzzuol, A. Montorsi, and L. Barbiero, which is under consideration for publication in SciPost Physics. I do not find the paper suitable for publication in its present form, because the text seems to lack accuracy. I suggest a major revision along the lines presented below.

==================================================================

In this theoretical and numerical study, the Authors analyse the Pair SuperFluid (PSF) Phase for two different Hamiltonians: (i) the single-species Bose-Hubbard model in 1D or 2D (Eq. 1) with on-site attraction and a three-body constraint; (ii) the two-species Bose-Hubbard model (Eq. 11) with on-site interspecies attraction and a hard-core constraint. They introduce a quantity (Eq. 8) which they call the 'odd parity', and show numerically that it is an order parameter for the PSF phase.

==================================================================

My two more important questions A, B are related to the odd parity defined in Eq. 8 on page 5:

(A) Could the Authors explain in greater detail how their quantity $O_P^{(0)}(j)$, defined in Eq. 8, provides different information from the closely related parity $O_P(j)$, previously defined in Ref. [10: Berg PRB 2008], and explicitly mentioned on p. 4, 3 lines from the bottom of the page ?

(B) The Authors write (abstract, l. 7; and again p. 2, 4 lines from end) that the odd parity is the "unique order parameter" of the PSF phase. However, Ref. [44: Bonnes PRL 2011] also introduce the order parameter <b_i^2> in a similar context. What are the differences in the roles of these two quantities ?

==================================================================

The following remarks (1-4) are more minor, but may help in making the manuscript more accessible to a non-specialised audience.

(1) Concerning the second considered model (Eq. 11): Figure 6a seems to suggest that the species A and B are trapped in two spatially-separated optical lattices, but this does not explicitly appear in the Hamiltonian: does this subtlety in the proposed realisation play a role ? Why do the paired bosons belong to sites of the two lattices which differ by one lattice spacing rather than to corresponding sites ?

(2) The Authors use two different methods: bosonisation (Sec. 2.1) and DMRG (Sec. 3). Matrix Product States are also mentioned once in Sec. 3.2. Would the Authors consider including a section entitled "Methods", in which they discuss the strengths and limitations of these approaches ?

(3) Two keywords seem to be used without a definition or reference: (a) "snake-like path", on p. 7, paragraph 1, line 6; (b) "C_4 symmetric 2D limit", on p. 7, paragraph 3, line 2.

(4) I am surprised at the Authors' usage of "normal superfluid" as a single keyword (abstract, l. 13; and again p. 2, paragraph 3, l. 9). Is this standard terminology? If so, could the Authors provide a reference using it?

Recommendation

Ask for major revision

---

## Round 2 · Referee Report · Masaki Oshikawa (Referee 3) · 2024-7-1

Report

This paper proposes a non-local order parameter "odd parity", which is a modified version of "parity" operator introduced in earlier literature. The authors argue that it characterizes the pair superfluid (PSF) phase, and demonstrate numerically in 1 and 2 dimensions. Indeed the numerical results seem to confirm the picture, and the odd parity operator looks quite interesting and useful.

However, I find the discussion in the paper rather unsatisfactory and cannot recommend publication in the present form.

  1. For simplicity let us discuss 1 dimension. By definition, for a fixed string length $j$, the parity operator $O_p(j)$ and the odd parity operator $O_p^{(o)}(j)$ are related as

    $$ O_p(j) = (-1)^{j-1} O_p^{(o)}(j) $$
    . Then how can they detect different phases? I suppose that, the hidden assumption is that the nonlocal order parameter has a definite sign, i.e. it does not change the sign for sufficiently large $j$. So if $O_p(j)$ is non-vanishing but is oscillating and changes sign like $(-1)^j$, we do not regard the parity to be long-ranged but the odd parity is long-ranged (and vice versa). Do you agree? In any case, please clarify. It is also very important to describe how the authors extracted the non-local order parameter $O_p^{(o)}$ from the numerical data $O_p^{(o)}(j)$ for various different $j$'s. This should be one of the essential information the authors are obliged to present.

  2. As the authors note, under the three-body constraint, the Bose-Hubbard is mapped to the S=1 system. In the 1D context (S=1 chain), in my understanding, the PSF is nothing but the "XY2" phase identified by Schulz in the seminal paper Ref. [64]. Although the authors do cite Ref. [64], I believe that it would be fair to mention explicitly that the 1D PSF phase was essentially discovered in Ref. [64].

  3. Related to the previous point, in Ref. [64] the Antiferromagnetic (Neel) phase was also identified. I believe that the odd parity operator is also non-vanishing in the antiferromagnetic phase, which corresponds to a charge-density wave state in the Bose-Hubbard context. So the non-vanishing odd parity alone does not imply that the system is in PSF phase. For 1D Bose-Hubbard model, according to Fig. 2(b), the authors numerically confirmed the power-law decay of $D(r)$. Moreover, Fig. 3 (b) indicates that $\Delta_2$ always vanishes for the range of $U/t$ studied. So I agree that the system is indeed in the PSF phase when the odd parity is non-vanishing within the model studied by the authors. (However, one should also be able to find the antiferromagnetic phase by modifying the Hamiltonian.) For 2D, Fig. 4(b) might suggest that the system is PSF and not antiferromagnetic/CDW, but I do not feel the evidence is sufficient. How does $D(r)$ look like, for example at $U=-20$??

  4. Please give the detailed derivation of Eq. (9) (bosonized expression of the odd parity operator), which is rather important for this paper.

  5. I have mostly considered the simplest case especially in 1D. Please also carefully re-examine the cases of the arbitrary filling and the two-species system, following the above comments.

After I formed my own opinion, I also checked the reports by other referees; I think the concern of Referee 2 overlaps my points, especially (1) above.

Recommendation

Ask for major revision

---

## Round 3 · Referee Report · Anonymous (Referee 1) · 2024-8-22

Strengths

Brilliant paper

Weaknesses

No weaknesses

Report

The authors answered all queries. The paper is ready to be published

Requested changes

none

Recommendation

Publish (surpasses expectations and criteria for this Journal; among top 10%)

---

## Round 3 · Referee Report · Anonymous (Referee 2) · 2024-8-24

Report

This is my second reviewer's report on this manuscript. I stand by the conclusion of my previous report: In its present form, the considered manuscript does not seem suitable for publication in SciPost. A substantial revision may change this.

The following three comments, all present in my previous report, still apply to the revised version:

  1. The 'odd parity' order parameter, which the Authors put forward as their key contribution in this paper, should be defined more carefully, including an explicit discussion of the mechanism whereby either 'parity' or 'odd parity' may be finite while the other is not.

  2. The Authors' claim that 'odd parity' is the "unique order parameter" for pair superfluids (abstract, l. 7; page 2, 3 lines from end) should be justified, by comparing it to previously proposed order parameters for pair superfluids, and explicitly giving the assumptions which make them inapplicable in the case considered by the Authors.

  3. The Authors' presentation is quite technical. Including an overview of the roles of the employed approaches would make it more accessible.

Recommendation

Ask for major revision

---

## Round 3 · Referee Report · Masaki Oshikawa (Referee 3) · 2024-8-28

Report

As I indicated in the report on the previous version, this paper contains interesting results. The new version of the paper is also much improved from the previous version. However I am still not satisfied with the lack of the details on the estimate of the odd parity from numerical data.

Requested changes

I presume that the odd parity is numerically estimated from an extrapolation of finite-size numerical data to infinity, but the details are lacking in the paper. It looks rather strange that the authors show the size dependence explicitly for the gap in Figure 3 but not for the odd parity, which is the main subject of the paper. There is a brief cryptic remark just before Section 3.1 -- "the nonlocal order parameter $O^{(o)}_P$ has been extracted from $O^{(o)}_P(j)$ by fixing $j$ to a sufficiently large value.... (i.e. not close to the transition)", but this description is not sufficient. First, the authors should demonstrate the point explicitly since this is the central point of the paper. Furthermore, it seems like the authors actually study the parameter $U/t$ quite close to the critical point in Fig. 2(c). Are they still "not close to the transition" concerning the odd parity?? (It would be rather surprising if that is the case.) In any case, the authors should show the size dependence of the odd parity explicitly for Fig. 2(c), Fig. 4(c), Fig. 5, and Fig. 6.

Recommendation

Ask for major revision

---

## Round 3 · Author Response

\documentclass{article}
\usepackage{graphicx}
\usepackage{amsmath}
\usepackage[margin=1in]{geometry}
\usepackage{xcolor}
\usepackage{comment}
\usepackage{hyperref}
\usepackage{cite}

\title{Reply Paper Scipost}
\author{Nitya Cuzzuol, Arianna Montorsi and Luca Barbiero}
\date{July 2024}

\begin{document}

\maketitle

\section{Report1}
\textit{it would be useful to core a review on non-standard Hubbard models where pair supefluidity is discussed thoroughly in various context:}\\
\begin{itemize}
\item 1. Omjyoti Dutta, Mariusz Gajda, Philipp Hauke, Maciej Lewenstein, Dirk-Sören Lühmann, Boris A. Malomed, Tomasz Sowiński, Jakub Zakrzewski, Non-standard Hubbard models in optical lattices, Rep. Prog. Phys. 78, 066001 (2015), arXiv:1406.0181.
\item 2. Titas Chanda, Luca Barbiero, Maciej Lewenstein, Manfred J. Mark, and Jakub Zakrzewski, Recent progress on quantum simulations of non-standard Bose-Hubbard modes, arXiv:2405.07775.
\end{itemize}
We thank the referee for the excellent comments and evaluation of our paper, and for recommending it for publication. Following the suggestion, we introduced the citation to both reviews in the introduction of our paper.

\section{Report2}
\textit{I do not find the paper suitable for publication in its present form, because the text seems to lack accuracy. I suggest a major revision along the lines presented below.}\\
We thank the referee for carefully reading our manuscript. From his/her observations we understand that some of the known results obtained in previous literature could be further detailed in the present manuscript to render its reading more self contained. We appreciate the suggestion.
\begin{itemize}
\item (A) \textit{Could the Authors explain in greater detail how their quantity $O_P^{(o)}(j)$, defined in Eq. 8, provides different information from the closely related parity $O_P^{(o)}$, previously defined in Ref. [10:Berg PRB 2008], and explicitly mentioned on p. 4, 3 lines from the bottom of the page ?}\\
Parity and odd parity are related by
\begin{equation}
O_P(j) = (-1)^{j-1}O_P^{(o)}(j)
\end{equation}
As explained in more details in Ref.[11] for the very same relation holding between charge and spin parity in fermionic case, the finiteness of the expectation value of the uniform part of one or the other captures phases with different orders. In particular, $O_P$ turns out to have a finite uniform part $(O_P(j)+O_P(j+1))/2$ in the MI phase, in which the uniform part of $O_P^{(o)}$ is vanishing. Instead the latter becomes finite in the PSF phase, in which the uniform part of $O_P$ goes to zero. We added a sentence on this point in the revised manuscript.
\item (B) \textit{The Authors write (abstract, l. 7; and again p. 2, 4 lines from end) that the odd parity is the "unique order parameter" of the PSF phase. However, Ref. [46: Bonnes PRL 2011] also introduce the order parameter $\langle b_i^2 \rangle$ in a similar context. What are the differences in the roles of these two quantities ?}\\
First of all, in the present context in which the total number of particle is conserved, $\langle b_i^2 \rangle$ cannot be an order parameter. Indeed, $b_i^2$ correlations in PSF phase decay to zero algebraically in the thermodynamic limit [44], unlike our order parameter which correlations remain finite, as expected for a true order parameter. Secondly, while such algebraic decay for $b_i^2$ correlations is observed in both the SF and the PSF phases, $O_P^{(o)}$ is vanishing in the superfluid phase and becomes finite only when entering the paired superfluid phase.
\end{itemize}
\textit{The following remarks (1-4) are more minor, but may help in making the manuscript more accessible to a non-specialised audience.}
\begin{itemize}
\item 1) \textit{Concerning the second considered model (Eq. 11): Figure 6a seems to suggest that the species A and B are trapped in two spatially-separated optical lattices, but this does not explicitly appear in the Hamiltonian:}
\begin{equation}
H = - t \sum_{i, \sigma = \rm {A,B}} (b_{i,\sigma}^{\dagger} b_{i+1,\sigma} + \text{h.c.}) + U \sum_i n_{i,\rm {A}} n_{i,\rm {B}} %+ V \sum_{i,\sigma=A,B} n_{i,\sigma}n_{i+1,\sigma}
\end{equation}
\textit{does this subtlety in the proposed realisation play a role ?}\\
The model we are analyzing here can be described as a couple of chains of hard-core bosons or equivalently as a single chain populated by two species of hard-core bosons. $t$ is the hopping coefficient of hard core bosons and there is no interchain/interspecies hopping. The interaction between the two species/two chains is regulated by $U$. In order to further clarify the notation we added a comment on the index $i$ in the text.\\

\textit{Why do the paired bosons belong to sites of the two lattices which differ by one lattice spacing rather than to corresponding sites ?}\\
A pair of atoms on different chains and same site index corresponds to the double occupied site of the constrained BH model. The two atoms displaced by one lattice site in the two chains are representative of a broken pair maintaining a finite correlation length. In fig.6a the red ellipses highlight such finite correlation length for the two single bosons, which is consistent with finite odd parity.\\
We have added a comment on this point in the caption of fig.6.
\item 2) \textit{The Authors use two different methods: bosonisation (Sec. 2.1) and DMRG (Sec. 3). Matrix Product States are also mentioned once in Sec. 3.2. Would the Authors consider including a section entitled "Methods", in which they discuss the strengths and limitations of these approaches ?}\\
Since they are both standard methods in the context of low dimensional strongly correlated systems, we don't feel that an introduction in this context is appropriate. Notable examples of Bosonization analysis can be found in \cite{Schulz_1986, Berg_2008, DallaTorre_2006, Timonen_1985, Luther_1977, Orignac_1998, Shelton_1996, Nersesyan_2020, Nakamura_2003}. While DMRG results in \cite{Kuhner_1998, Kuhner_2000, DallaTorre_2006, Berg_2008, Ejima_2015, Ejima_2021, Rossini_2012, Dalmonte_2011, Diehl_2010, DegliEspostiBoschi_2003, DegliEspostiBoschi_2016, Fraxanet_2022}.
\item 3) \textit{Two keywords seem to be used without a definition or reference:}
\begin{itemize}
\item (a) \textit{"snake-like path", on p. 7, paragraph 1, line 6;}\\
This keyword means the way one enumerate the sites of a 2D lattice to obtain a single chain. It can be found in the references cited for the TeNPy libraries Ref.[69]. We will add the citation after this keyword.
\item (b) \textit{"$C_4$ symmetric 2D limit", on p. 7, paragraph 3, line 2.}\\
We are referring here to the rotation symmetry of the square lattice by $\pi/2$ rotations. However, we decided to remove this sentence to avoid any confusion.
\end{itemize}
\item 4) \textit{ I am surprised at the Authors' usage of "normal superfluid" as a single keyword (abstract, l. 13; and again p. 2, paragraph 3, l. 9). Is this standard terminology? If so, could the Authors provide a reference using it?}\\
The authors thank for the suggestion and will replace "normal superfluid" with "atomic superfluid" as in Ref.[43, 46]. %\cite{Diehl_2010, Bonnes_2011}. %[41,44].

\end{itemize}

\section{Report3}
\textit{This paper proposes a non-local order parameter "odd parity", which is a modified version of "parity" operator introduced in earlier literature. The authors argue that it characterizes the pair superfluid (PSF) phase, and demonstrate numerically in 1 and 2 dimensions. Indeed the numerical results seem to confirm the picture, and the odd parity operator looks quite interesting and useful.}\\
We thank the referee for the thorough reading of the paper and the valuable observations and suggestions provided. Following the suggestions and addressing the questions raised, we have integrated and updated the paper as discussed below.
\begin{itemize}
\item 1) Parity and odd parity are related as
\begin{equation}
O_P(j) = (-1)^{j-1}O_P^{(o)}(j)
\end{equation}
\textit{Then how can they detect different phases? I suppose that, the hidden assumption is that the nonlocal order parameter has a definite sign, i.e. it does not change the sign for sufficiently large $j$. So if $O_P(j)$ is non-vanishing but is oscillating and changes sign like $(-1)^j$, we do not regard the parity to be long-ranged but the odd parity is long-ranged (and vice versa). Do you agree? In any case, please clarify.}\\
This is the point. One should look at the asymptotic behavior of the expectation value of the uniform part of the two operators. Indeed the uniform part of $O_P(j)$, namely $(O_P(j)+O_P(j+1))/2$, has vanishing expectation value in the asymptotic limit in the pair superfluid phase, unlike that of $O_P^{(o)}(j)$ . Ultimately, this is rooted in the spin-charge separation of the bosonized Hamiltonian, and is perfectly analogous to the connection between spin and charge parity discussed in Ref.[11]. We added a comment on that along the manuscript.\\

\textit{It is also very important to describe how the authors extracted the non-local order parameter $O_P^{(o)}$ from the numerical data $O_P^{(o)}(j)$ for various different $j$'s. This should be one of the essential information the authors are obliged to present.}\\
The nonlocal order parameter $O_P^{(o)}$ has been extracted from $O_P^{(o)}(j)$ by fixing $j$ to a sufficiently large value to ensure that, within the region of parameter space where the model exhibits the phase defined by this parameter (i.e., not close to the transition), the nonlocal order parameter has already converged to a finite constant. We verified that in the same region the parameter $O_P(j)$ has an oscillating behaviour with vanishing average.
\item 2) \textit{As the authors note, under the three-body constraint, the Bose-Hubbard is mapped to the S=1 system. In the 1D context (S=1 chain), in my understanding, the PSF is nothing but the "XY2" phase identified by Schulz in the seminal paper Ref. [55]. Although the authors do cite Ref. [55], I believe that it would be fair to mention explicitly that the 1D PSF phase was essentially discovered in Ref.[55].}\\
The authors thank for the suggestion. We have added the information.
\item 3) \textit{Related to the previous point, in Ref. [55] the Antiferromagnetic (Neel) phase was also identified. I believe that the odd parity operator is also non-vanishing in the antiferromagnetic phase, which corresponds to a charge-density wave state in the Bose-Hubbard context. So the non-vanishing odd parity alone does not imply that the system is in PSF phase. For 1D Bose-Hubbard model, according to Fig. 2(b), the authors numerically confirmed the power-law decay of $D(r)$. Moreover, Fig. 3 (b) indicates that $\Delta_2$ always vanishes for the range of $U/t$ studied. So I agree that the system is indeed in the PSF phase when the odd parity is non-vanishing within the model studied by the authors. (However, one should also be able to find the antiferromagnetic phase by modifying the Hamiltonian.)}\\
\begin{figure}[ht]
%
\centering
\includegraphics[scale=1]{decayNN.png}
%
\caption{\label{NN}
Analysis of $(-1)^{|i-j|}\times\langle (n_i-\Bar{n})(n_j-\Bar{n}) \rangle$ for different values of the onsite interaction $U$ and nearest neighbour interaction $V$ for 1D and 2D. The correlator has long range order only for finite positive $V$ corresponding to CDW \cite{Cuzzuol_2024}.}
%
\end{figure}
The intuition that the odd parity is different from zero also in the charge density wave phase equivalent to the cited antiferromagnetic phase is correct. Indeed, in ref.\cite{Cuzzuol_2024} this was observed for the extended BH model for positive values of nearest neighbour interaction. However, contrary to PSF, the CDW phase is SSB ordered phase, and, besides being characterized by a finite value of both string and odd parity nonlocal orders, can also be characterized by the local order parameter $(-1)^{|i-j|}\times\langle (n_i-\Bar{n})(n_j-\Bar{n}) \rangle$ \cite{Baldelli_2024}, where $n$ is the number operator and $\Bar{n}$ the filling. This local order parameter has long range order only in CDW as represented in fig.\ref{NN}.\\

\textit{For 2D, Fig. 4(b) might suggest that the system is PSF and not antiferromagnetic/CDW, but I do not feel the evidence is sufficient. How does $D(r)$ look like, for example at $U= - 20$ ?}\\
Also for more negative $U$ values $D(r)$ is observed to have the same power law decay. For clarity we reported in Fig.(\ref{Dr}) the decay of $D(r)$ for $U = -17, -18, -20$. As a counterproof, in 1D the transition to CDW state was observed in \cite{Cuzzuol_2024} only for positive values of nn interactions. Moreover one can verify that $(-1)^{|i-j|}\times\langle (n_i-\Bar{n})(n_j-\Bar{n}) \rangle$ doesn't have long range order for negative $U$ in Fig.\ref{NN}.
\begin{figure}[ht]
%
\centering
\includegraphics[scale=1]{decaybd2b2_N6legs.png}
%
\caption{\label{Dr}
Analysis of $D(r)$ as defined in the paper for $U = -17, -18, -20$. For all three values of interaction $D(r)$ is always decaying power law.}
%
\end{figure}

\item \textit{4) Please give the detailed derivation of Eq. (9) (bosonized expression of the odd parity operator), which is rather important for this paper.}\\
A detailed derivation has been added to the section on bosonization, together with appropriate references to the literature on the related derivation for spin parity.

\item 5) \textit{I have mostly considered the simplest case especially in 1D. Please also carefully re-examine the cases of the arbitrary filling and the two-species system, following the above comments.}\\
Yes, we have also added a comment on the 2D limit.
\end{itemize}

\bibliographystyle{iopart-num.bst}
\bibliography{myRef_reply.bib}
\end{document}

---

## Round 3 · List of Changes

In response to the reviewers' comments and suggestions, we have updated the manuscript with the following changes: - Throughout the paper: we replaced "normal superfluid" with "atomic superfluid". - In the first paragraph of the introduction (line 27): we introduced "bosonic atoms in optical lattices [31, 32]" and cited the reviews suggested by Reviewer 1. - In the third paragraph of the introduction (line 42): we added "and corresponds to the XY2 phase in spin models [44, 55, 56]" to emphasize the correspondence between PSF and XY2 phases, as suggested by Reviewer 3. - In Section 2.1 (Bosonization analysis), starting from line 113: we made several changes to clarify this section and address the reviewers' questions. - In Section 3, before the beginning of Section 3.1: we introduced a new paragraph at line 168 to answer a question from Reviewer 3. - In Section 3.4, below Eq. (11) (line 235): to address a question from Reviewer 2, we specified that "denotes the generic site of each" for the i-index in Eq. (11). Additionally, in response to the same question, we updated the caption of Figure 6 on page 9.

---

## Round 4 · Referee Report · Anonymous (Referee 2) · 2024-10-15

Report

This is my third reviewer's report on this manuscript. I recommend
swift publication of the manuscript in its present state, in SciPost
Physics Core.

For details, please read below.

I recommend publication because this theoretical analysis is timely.
Indeed, the pair superfluid phases analysed by the Authors have
recently been predicted to occur in two-species lattice systems
involving either bosons or fermions (cf. the Authors' Ref. [64:
Grémaud & Batrouni PRL 2021]). In this context, the Authors'
theoretical work, involving an order parameter characterising these
phases in a situation which is both closer to projected experiments
and accessible with well-established numerical methods, is welcome.

I suggest SciPost Physics Core as the appropriate venue for two
reasons, both previously identified in my two previous reports.
Firstly, the Authors' presentation is quite technical. Secondly, the
comparison with previously introduced order parameters and their
validity conditions remains limited.

Recommendation

Accept in alternative Journal (see Report)

---

## Round 4 · Referee Report · Masaki Oshikawa (Referee 3) · 2024-10-30

Report

As I wrote in the earlier reports, the findings in the paper are interesting enough to warrant publication. However, I am rather perplexed by the apparent stubbornness of the authors to avoid including details in the paper. I did not mean I was doubting the authors' results or anything, but "the devil is in the details" especially in physics --- don't you agree?
I do not really want to prolong the refereeing process any more.
However, please, please, for the sake of the community, include the nice Figure 1 in the "Authors comments upon resubmission" to the paper. Actually I would like to see similar analysis for other data, but since you have already made the plot, it should not be too difficult or time-consuming to include at least that one figure into the paper.
If the authors comply to this point, I do not need to review the paper again.

Recommendation

Ask for minor revision

---

## Round 4 · List of Changes

• In the Abstract (line 8): we added the line ", in regimes of total particle number conservation".
  • End of section 1: we added the lines from 59 to 64, including a new reference (57).
  • End of section 2.1 (below eq.9): we modified lines 139-143.
  • Section 3: we added the lines 161-162.
  • Section 3.1: we modified lines 184-185.
  • caption figure 3: we added " for finite DMRG".
  • caption figure 4: we added "by iDMRG simulations".
  • caption figure 6: we added "performing iDMRG simulations with a maximum bond dimension of χmax = 400."

---

## Editorial Decision

published